# A Model of Factors Determining Nurses’ Attitudes towards Learning Communicative Competences

**DOI:** 10.3390/ijerph18041544

**Published:** 2021-02-05

**Authors:** Lucyna Iwanow, Mariusz Jaworski, Joanna Gotlib, Mariusz Panczyk

**Affiliations:** Department of Education and Research in Health Sciences, Faculty of Health Science, Medical University of Warsaw, 02-091 Warsaw, Poland; lucyna.iwanow@wum.edu.pl (L.I.); mariusz.jaworski@wum.edu.pl (M.J.); joanna.gotlib@wum.edu.pl (J.G.)

**Keywords:** registered nurses, social skills, interpersonal relations, communicative competences, postgraduate education, empathy, working environment

## Abstract

The aim of the study was to assess the empirical model of factors determining the attitude towards acquiring communicative competences among nurses participating in the program of specialist training courses. Research was conducted using a cross-sectional study. A representative group of 1010 Polish registered nurses that took part in the postgraduate education course answered a self-report survey (three instruments: NEO-PI-R questionnaire, Communication Skills Attitude Scale (CSAS), and Empathy Understanding Questionnaire (KRE II)) from the beginning of March to the end of May 2018, which was evaluated using path analysis. The research results conducted confirmed the soundness of the created theoretical model (χ^2^ = 0.278, *p* = 0.598, RMSEA < 0.05). It was proved that acquiring communicative competences in nurses is determined by factors such as professional experience, empathic tendency, and the intensity of agreeableness, whereby these factors are bound with each other creating a homogeneous network. The developed model demonstrated that skills can most effectively be shaped in an individual’s attitude based on positive mentoring in work environment.

## 1. Introduction

Nurses’ communicative skills influence the quality and efficiency of the therapeutic process as well as the safety of both the patient and medical personnel in a significant way [1]. Effective communication in the nurse-patient relationship is the key to shortening the duration of hospitalization, increasing patients’ involvement in the therapeutic process, and the quality of care provided [2,3]. Despite the important role of these skills in clinical practice, insufficient level of these skills among nurses is still stressed in the literature, as well as a large number of iatrogenic mistakes connected with them [4]. This is most likely to be related to the lack of shaping a positive attitude towards these competences. Simply having knowledge in this area is thought not to condition taking appropriate steps. In other words, a nurse may have knowledge concerning the role of communicative skills in clinical practice; however, she does not have to apply this knowledge as her attitude maybe a negative one. Due to this fact, it is essential not only to pass on knowledge, but also to shape appropriate attitudes at every level of education [2,5]. It should be noted that an attitude is not a constant and invariable human disposition. It may be subject to modifications under the influence of external and internal factors. Thus, it is particularly important to become familiar with factors which may modify it to the highest degree [6,7].

In the case of external factors, the work environment can be most important. A nurse spends a greater part of her day at work. Therefore, this is the environment that affects not only her well-being and life satisfaction, but also shapes and modifies her attitude towards patients, co-workers, and medical procedures [8]. Numerous studies stress the relationship between professional experience or the study year and the attitude towards acquiring communicative competences [9,10,11,12,13]. The function of professional experience in shaping attitudes towards acquiring communicative competences is not clear. There is research indicating that professional experience increasingly influences more positive attitudes [13,14], in others—the contrary [15]. Other publications indicate that professional experience does not play a statistically significant function in creating attitudes [16]. The reasons for these discrepancies may be found in four areas [13,14,15]. Firstly, shaping a positive attitude is labor-intensive and requires great personal engagement, therefore one’s own work is of importance here. Due to that fact, such a person must not only be aware of the need to acquire competences, but also should reflect on their own skills. Nurses may present a different level of self-work in the context of improving their own skills despite having the same work experience. Secondly, it is thought that affective attitude is most susceptible to modification resulting from external factors, particularly when it is related to experiencing strong emotions. This requires the skill of being able to cope with emotions that are not only our own, but also those of other people. Therefore, what should be included is the aspect of emotions, and it should be done by analyzing the shaping of a positive attitude towards acquiring communication skills. Thirdly, while shaping a positive attitude it is crucial to bear in mind personal contact with a patient, i.e., experience. It is emphasized that knowledge itself is not sufficient. Despite the fact that at the onset it might translate into a positive attitude towards acquiring communication skills, it diminishes in the long run when not supported by experience. This is why the character of the working environment in which a nurse works is important. Fourthly, shaping a positive attitude ought to include long-term training in communication skills that would begin early and last throughout all years of education [13,14,17].

Bearing in mind the results of the studies, it must be stressed that a simple assumption of the relationship between professional experience and shaping positive attitude towards communicative skills is significantly flawed. Professional experience is directly connected with work environment and the skill of dealing with it. According to Raghubir [18] empathic tendency, especially high levels of emotional intelligence, enables coping efficiently in a stressful and emotionally burdensome work environment, such as, undoubtedly, clinical work. It is assumed that the skill of understanding and identification of one’s own and others’ emotions is an important factor influencing adequate decision-making in clinical practice [19], thus increasing the efficiency of patient engagement in the therapeutic process, improving therapeutic relationships, and positively influencing the quality of medical care offered to the patient [18]. It is worth stressing that the ability to work with one’s own and others’ emotions may directly influence the gaining of professional clinical skills in nurses [20], as well as constituting a significant factor in successful nursing practice [21]. This thereby points to the relationship between professional experience and empathic tendency. It may be assumed that empathic tendency, connected with the skill of identifying and managing emotions, should also influence the shaping of positive attitudes towards acquiring communicative competences.

Among the internal factors, personal traits should be of particular importance. It is precisely the nurses’ personalities that may condition their perseverance in mastering their own skills through shaping a positive attitude towards acquiring them [22]. Personality traits are relatively constant tendencies conditioning human behavior in various situations. Therefore, they may constitute the foundation for shaping all attitudes and psychosocial skills. This relation has been highlighted in some studies. Ayuso-Murillo et al. [23], among others, as well as Chan and Sy [24] paid attention to this link. One of the most known approaches to personality in the literature is the Big Five Personality Traits devised by Costa and McCrae [25]. The Big Five presumes that all people possess five basic traits (neuroticism, extraversion, openness to experience, agreeableness and conscientiousness), but they differ among one another with the intensity of each of the traits [25]. The complexity and variety of personality traits as well as their mutual interactions pose a difficulty in analyzing all the traits at the same time. Due to this fact, it is essential to determine those traits that would have the greatest impact on shaping a positive attitude towards acquiring communication skills. According to Chan and Sy [24] agreeableness, which conditions human behavior in interpersonal contacts, may be of key importance. It is a feature that is expressed through many types of human behaviors. It should be stressed that agreeableness is a complex personality trait and comprises of several subdimensions [25]. Therefore, it is not a uniform feature. One of its sub-dimensions is compliance, which is thought to call for special attention in the context of nursing [26]. This is related to the characteristics of the feature itself, since it concerns interpersonal relations and communication skills [25], i.e., two skills that are important in the clinical work of a nurse [27]. People with a high level of this trait have the need to positively solve any conflict at interpersonal level [25]. It should be emphasized that every person has this trait and individual differences only concern its intensification, which can translate to the awareness of improving one’s communication skills. Such awareness affects the feeling of psychological discomfort and shapes the need for change and modification of one’s own skills [26]. Thus, it may be assumed that agreeableness may also be of key importance in the context of nurses’ attitudes towards learning communicative skills. This, however, demands empirical verification.

As can be seen from the above, shaping a positive attitude towards acquiring communication skills is a complex process that undergoes modification. It is thought that activities in this area should be carefully planned [14]. Such actions may manifest themselves in various activities, e.g., courses, mentoring, and postgraduate training. By virtue of its nature, which combines professional experience and boosts nurses’ expertise, post-graduate training provides ideal conditions for positive reinforcement of attitudes towards learning communicative competence. What is more, it also provides a safe area for facing one’s doubts relating to practical applications of acquired skills and encourages reflection. It seems that the role of postgraduate education in this area is particularly important.

Postgraduate education is diversified in both EU member countries and worldwide [28]. In the majority of countries, nurse specializations form part of this stage of education in given fields of medicine/nursing (geriatrics, surgery, anesthesiology, pediatrics, etc.) [28,29,30]. The aim of postgraduate education is to improve skills already acquired in the course of university studies, and their development in order to gain new competences. As a result of current trends, as well as alarming reports on the deficit of soft skills among medical personnel, components pertaining to communication were included in the educational profile in the current programs of courses and trainings [31,32,33].

The Act of 15 July 2011 on the professions of nursing and midwifery regulates postgraduate education of nurses and midwives in Poland. In line with the Act, “Nurses and midwives have the duty to constantly update their knowledge and professional skills and the right to exercise professional development in various kinds of postgraduate education” [34]. Polish legislation provides for various forms of postgraduate education, but one of the most important forms is specialist training [34].

The aim of this study was to assess an empirical model of factors determining the attitude towards learning communicative competences among nurses participating in the program of specialist training courses.

## 2. Materials and Methods

### 2.1. Design

A cross-sectional study administered to a representative group of Polish registered nurses was carried out from the beginning of March to the end of May 2018.

### 2.2. Sample Size

The size of sample needed for the study was calculated employing structural equation modelling (SEM) [35]. The number of observed variables was assumed at the level of *n* = 4 and number of unobserved variables at the level of *n* = 3. The predicted effect size was 0.1. The statistical power (0.80) and probability (0.05) levels were assumed a priori. With such assumptions, the minimum sample size required to detect the specified effect (*n* = 400) was determined, as was the minimum sample size required given the structural complexity of the model (*n* = 950). It was assessed that the final size of the required sample should total no fewer than 1000 cases.

### 2.3. Participants

The main criterion for inclusion was participation in (with positive outcome) the program of specialist training held in the years 2016–2017. An additional criterion for inclusion was at least three years of documented professional experience in the profession of a nurse. Potential participants of the study were recruited via a database compiled by the Centre for Postgraduate Education for Nurses and Midwives in Warsaw, Poland. A mailing list of potential participants meeting the inclusion criteria was created. The person completing the online questionnaire did not provide personal data that would enable identification in the database. No data were recorded to link the e-mail address of the participant with the data from the online questionnaire.

The sampling frame involved the register of 17,500 nurses with the title of a specialism obtained in the years 2016–2017. Using a simple random sampling method (computer generated random numbers), a sample of 4000 people was drawn. In the study, 2030 people (response rate 50.8%) agreed to participate in the study, and a data package was received from 1010 nurses.

### 2.4. Data Collection

People randomly selected to participate in the study were informed about the aim of the study, as well as about the means of data collection and storage. Informed consent was obtained verbally from each participant from whom data were collected after filling in the questionnaire. There was no register of people who did not agree to participate in the study. The reasons of refusal were not recorded. Questionnaires were anonymous and after data collection there was no possibility of identification of personal data of the participant filling in a given questionnaire.

The results of the study were collected using a computer-assisted web interviewing (CAWI) method. An online questionnaire was developed in a program for creating web interviews. People who agreed to participate in the study were provided with a link to the online questionnaire. No information was recorded about the personal data of the study participants.

### 2.5. Validity, Reliability and Rigour

The instruments used in this study were checked and tested on senior Polish nurses [36,37,38].

### 2.6. Instruments

In order to measure the intensity of agreeableness, the subscale ‘agreeableness’ from the questionnaire NEO-Pi-R (Revised NEO Personality Inventory) by Costa and McCrae [25] in the Polish adaptation by Siuta [37] was employed. The subscale is built of 8 statements, the truthfulness of which is rated on a 5-point scale by the respondent, where 1 stands for “strongly disagree”, and 5—“strongly agree”. Total level of agreeableness is the sum of points converted in accordance with a key developed by the authors. The measurement of the variable was conducted by a psychologist. NEO-PI-R is a fair and relevant tool, commonly used for studying personality.

The standardized questionnaire Communication Skills Attitude Scale (CSAS) by Rees, Sheard [39], in the Polish adaptation by Panczyk et al. [36], was used for the measurement of attitudes towards learning communicative competences. The Polish version of the questionnaire is made of 23 statements pertaining to attitudes towards learning communicative competences in the teaching profession and in professional practice. Statements were assessed on a five-degree Likert scale where 1 means “strongly disagree” and 5—“strongly agree”. The tool is divided into two subscales: positive (PAS) and negative (NAS). The positive subscale is made up of 11 statements, and the negative subscale has 12 statements. The evaluation of attitudes towards learning communicative competences is done on the basis of the total result of the scale, after recalculation of the score in accordance with the described key [36].

The Polish Empathy Understanding Questionnaire (KRE II) by Węgliński [38] served for the measurement of empathy, understood both as the emotional and cognitive component, as well as to capture the meaning of motivational aspect in empathy. The questionnaire is made up of 33 statements, assessed on a 4-point scale. The questions are divided into ones where the diagnostic measurement is positive “yes” and those of negative diagnostic measurement “no”. The questions with the power “yes” are evaluated on a four-degree scale, where 3 stands for—yes, 0—no; while questions with the power “no”, on the reverse scale, 3 stands for—no, and 0—yes. The tool is divided into 5 subscales: (1) sympathizing with the pleasant and unpleasant experiences of others; (2) co-feeling pleasant and unpleasant experiences with others; (3) sensitivity to the experiences of others; (4) readiness to sacrifice for others; (5) empathizing in the conditions and experiences of others. The evaluation of empathic understanding is determined by means of the total result of the scale obtained in accordance with the guidelines provided with the questionnaire. Points are counted from each subscale—the maximum number of points which may be gained in each subscale is the product of the number 3 and the number of statements.

Data pertaining to professional experience measured in years were downloaded from the database Centre for Postgraduate Education for Nurses and Midwives in Warsaw, Poland. Consent for access to the data was obtained.

### 2.7. Theoretical Assumptions of the Model

Based on literature review, a theoretical model of potential factors determining nurses’ attitudes towards acquiring communicative skills was developed. Agreeableness, as a personality variable, was regarded as an observed exogenous variable. The discussed variable conditions represent the relatively stable dispositions of an individual to undertake specific activities, and do not undergo modifications. It may therefore be assumed that agreeableness will influence other observed endogenous variables (attitude towards acquiring communication skills, empathic tendency, and seniority) (Figure 1). Agreeableness may positively influence empathic tendency, as well as seniority. It is presumed that agreeableness as a psychological variable connected with the ability to work with others will influence the need to shape empathic tendency in nurses and condition adaptive mechanisms to work conditions, and thus support professional experience in a given profession. In addition, longer traineeships in clinical practice may be connected with shaping adequate adaptive mechanisms to work conditions (e.g., cooperation in a therapeutic team, the patient, and their family), health education of the patient, increase of treatment efficiency, and adherence to medical recommendations. This, in turn, may influence the need for greater insight into the patient’s needs, and show the illness from the patient’s perspective (a situation in which they found themselves). In order to achieve this, empathic tendency may be helpful.

### 2.8. Ethical Considerations

The authors sought advice from the Bioethics Committee of Medical University of Warsaw to conduct the presented study. As the “commission does not issue opinions onsurveys, retrospective and other non-invasive scientific studies”, approval was not required (AKBE/37/19).

### 2.9. Data Analysis

In order to determine causal links between three psychological variables and professional experience, the method of path analysis, i.e., SEM (Structural Equation Modelling), was employed. The aim of the analysis was to obtain answers to the questions whether causal relationships between variables presumed by the researcher in theory are confirmed in the collected empirical data. In order to do that, model parameters were assessed (path coefficient, variance, and covariance) which served as a theoretical foundation for the variance–covariance matrix for variables used in the model. Estimation of the parameters of the model consisted in a selection of the parameters such that the theoretical matrix of the variance-covariance was as close as possible to the observed matrix of variance-covariance. For the estimation of the structural model, the maximum likelihood method was used. The difference between the theoretical and empirical matrices was estimated based on three values: discrepancy function FMIN (minimum fit function), CMIN (chi-square statistics), and CMIN/DF (normal chi-square). For the evaluation of the model, the chi-square statistics were expected to be nonsignificant. As the CMIN value is dependent on the size of the sample and the degree of the complexity of the model, Hoelter’s critical N, also called the Hoelter index was also calculated. CMIN must be less than 0.5, and Hoelter’s critical N must be greater than 200 to show a good fit index.

SEM has no single statistical test that best describes the strength of the model prediction. Instead, different types of measures were developed by researchers in combination to assess the results. The commonly used fit indices in the literature include the related CMIN, Goodness of Fit Index (GFI), Adjusted Goodness of Fit Index (AGFI), Comparative Fit Index (CFI), Tukey-Lewis Index (TLI), Normed Fit Index (NFI), Incremental Fit Index (IFI), and Root Mean Square Error of Approximation (RMSEA). According to Byrne [35] GFI, AGFI, CFI, TLI, NFI, and IFI measures equal to or greater than 0.95 signify good fit indices. In addition RMSEA less than 0.05 displays the most acceptable fit index.

All statistical calculations were performed using the statistical package IBM^®^ SPSS^®^ Statistics (Chicago, IL, USA), v23 and Amos v21. For all analyses, a *p*-value of < 0.05 was considered statistically significant.

## 3. Results

### 3.1. Participant Characteristics

The average age of participants was 41.6 years (SD = 8.14, Min. = 26.0, Max. = 63.0), and average years spent at work in the profession was 18.5 (SD = 9.04, Min. = 3.0, Max. = 40.0). The sample was representative of the broader Polish nurse population in terms of the title of specialism that a nurse held. Table 1 reports demographic characteristics of the participants who completed this study.

### 3.2. Descriptive Statistics of Variables

The score for the obtained results for particular scales (agreeableness, empathic tendency, and attitude towards learning communication skills) were recalculated to the scale, in line with the published norms for the validated tools: NEO-Pi-R, KERII, and CSAS. All the variables included in the model of structural equations were analyzed for the presence of data outliers (Mahalanobis distance). It was also calculated to what degree parameters such as skewness and kurtosis deviate from the ones that are expected for normal distribution. Although multivariate of normality was not satisfied (critical area −2.54, multivariate of kurtosis −1.11), particular variables did not prove significant divergence in the scope of compliance with normal distribution, as skewness and kurtosis were in the range from −1.5 to +1.5 (Table 2).

### 3.3. Measurement Model

Each of the calculated measures (FMIN, CMIN, and CMIN/DF) indicates that the assumed theoretical model finds confirmation in the collected empirical data (Table 3). Besides, on the basis of the obtained probability test (*p* = 0.598) it was assumed that the hypothesis on lack of differences between theoretical and empirical variance-covariance matrices is highly probable. Furthermore, the estimated Hoelter index value (0.05 levels of significance, *n* = 13,940) indicates that the verification of the zero hypothesis should not be vitiated by an error stemming from the sample size and the degree of complexity of the model tested.

As the aim of testing has never been to bring its fit to empirical data coming only from the sample studied but also in relation from the whole population, both the value of the discrepancy function F0 and the value of RMSEA index adjusted for the number of degrees of freedom were calculated (Table 4). Based on the assessed value F0 it was stated that matrices of variance-covariance are equal to the population matrix. In addition, the RMSEA value (<0.05) indicates the equality of both matrices. The results confirm a good fit of data to the expected model structure.

For a more accurate evaluation of the degree of fit of the model to data collected, two groups of indices of fit were assigned. The first group of indices was of the ones that compare the tested model with the zero model, i.e., the model whose matrix variance-covariance equals zero. The GFI value was estimated (1.000) as well as its corrected value, AGFI (0.998). The results indicate that nearly 100% variability of the dependent variable is explained by the tested model.

In the other group of indices, there are ones which compare the model tested with an independent model, i.e., one in which all variables in the model are not correlated. Minimally acceptable values for the indices are 0.95 (Table 5).

### 3.4. Assessing the Measurement Model

The analysis of the proposed path model demonstrated that the strongest direct effect connected with the influence of independent variables on the dependent variable takes place between the variables Agreeableness and Empathy (0.274). However, the weakest direct relation was observed in the case of variables Agreeableness and Seniority (0.151). Among the analyzed relationships, all were positive, with the exception of one, Agreeableness --> Communication (−0.202). A detailed summary presenting estimation results of model parameters of structural equations is shown in Table 6.

In the model analysis, both direct effects and the value of indirect and total effects were calculated, for which the influence of independent variables on the dependent variable is not direct. Indirect effects in the model described were implemented in the following paths (standardized path values were provided):-Agreeableness → Empathic tendency →Attitude towards teaching communication skills (0.271×0.228 = 0.062)-Agreeableness → Seniority → Empathic tendency → Attitude towards teaching communication skills (0.151 × 0.224 × 0.228 = 0.008).

The total value obtained from the effects indicates that Agreeableness indirectly influences the variable Communication (0.062 + 0.008 = 0.07). This means that the total influence of Agreeableness on Communication equals (−0.202 + 0.07 = −0.132). All correlations of the tested model, together with evaluation of path, were presented in Figure 2.

## 4. Discussion

The developed theoretical model and its empirical verification demonstrated the existence of important factors shaping the nurses’ attitudes towards acquiring communicative skills. The elaborated model clearly emphasizes the relationship between the external and internal factors in the process of shaping a positive attitude towards acquiring communication skills by nurses. This shows the complexity of the process and creates space in which modification activities may be introduced. In the elaborated model, there are on the one hand, variables directly related to the psychological traits of a nurse that play an important role, whereas on the other hand, external factors play a major role. This model places a great emphasis on the need to see the problem of shaping a positive attitude towards acquiring communication skills from different viewpoints. Thus, the role of just one factor should not be overestimated; on the contrary, their mutual dependency should be underlined. In addition, it should be noted that factors which create that model may vary from each other, depending on the context of the range in which they are modified and depending on any change over time.

Features that may undergo significant modifications are empathic tendency and professional experience. While analyzing the role of empathic tendency in shaping nurses’ positive attitude towards acquiring interpersonal communicative competence, two significant factors must be identified. The first one, the notion of empathic tendency, is well grounded in the literature [40,41,42], which boosts the theoretical framework of the developed model. Secondly, empathic tendencies may undergo modifications, and thus be reinforced by trainings in the field of improving interpersonal skills [21,43,44]. The ability to work with emotions should be underlined here as it is key in shaping a positive attitude [45]. If an individual experiences strong emotions that she or he is unable to deal with, then such an individual begins to seek alternative ways that would help reduce them. Such activities will not always be related to acquiring new competences. They may be focused on denying or reducing the importance of these competences. This is confirmed in scientific works which clearly emphasize the fact that an affective attitude means being more sensitive to changes [2,46]. Thus, the elaborated model allows space in which education related to that area may take place. However, the role of empathic tendencies in shaping a positive attitude towards acquiring communications skills should not be underestimated. In the case of nurses who are characterized by high empathic tendencies, such activities may bring good results. For instance, this may enable deeper insight into oneself, self-reflection, and constructive criticism, which may get translated into shaping the perceived need and positive attitude towards acquiring interpersonal competencies. This positive attitude will in turn condition creating safe and friendly work environments, as well as relevant responses to the patient’s needs [33,47]. One needs to remember that emotions that are experienced by an individual change and undergo modifications under the influence of external factors [48]. Therefore, it would be a mistake to omit them or reduce their role in this model. For that reason, it is impossible to unequivocally assume that people with empathic tendencies would be more open to developing a positive attitude towards acquiring communication skills. Just the presence of an external factor that influences the emotional state of a person might suffice to modify this dependence in an individual.

Any contact with strong external factors is related to professional experience. In clinical work, a nurse experiences many situations that influence her well-being and shape relationships with a patient [5,8,10,49]. Stress is a strong factor related to the working environment of a nurse and it has an impact on her emotional state [50]. It is thought that the lack of capacity for stress management has behavioral, physical, and mental consequences. One of the methods that may help nurses to manage stress effectively, is placing emphasis on shaping a positive attitude towards acquiring communication skills [51]. It is particularly important to point to appropriate communication methods which would improve their functioning in the working environment, lower the impact of stress and effectively help manage it. It is also important to show the dependency between strong stress and an emotional state [52]. In this way, not only the necessary knowledge is delivered, but also a form of training is experienced. Thus, knowledge and experience are combined. However, it needs to be emphasized that the form of transferring knowledge should be adjusted. In some cases, self-reflection may prove useful, which would identify the need of acquiring new skills. In some other cases, however, modelling might prove effective, i.e., presenting what activities are most effective and efficient. When choosing a method, a nurse’s personality may be of the essence [53].

Strong external factors that negatively influence the emotional state of a nurse may results in the adoption of various responses, not necessarily adaptive (e.g., flight reaction as a response to patient’s expectations). The flight reaction may be connected with a personality trait such as agreeableness, one of elements of the Five Factor Model [25]. It is a trait representing attitudes to other people, that is in what way a human perceives the feelings of others and reacts to them. In the presented model, a negative correlation was observed between agreeableness and attitudes towards communicative competences. However, this negative impact should not be overestimated, as this personality trait does not function in isolation. It interacts with other psychological features. Thus, considering the existence of other psychological features, it may appear that the discussed influence is reduced or hidden. Personality traits do not undergo modification, therefore it is not possible to alter them. What is possible, however, is the change in expressing this personality trait in the presence of other features. It is crucial also to become aware of presenting a high or low intensification of this trait [25]. Thus, a nurse may evaluate whether her behavior is related to the external factors (e.g., stress) or inadequate skills.

The analysis of the trait seems justified in relation to empathic tendency and self-reflection. For example, a high level of this trait with simultaneous low level of empathic tendencies and self-reflection may negatively influence shaping of the attitude. In this model, the use of defense mechanism in the form of flight/avoidance may be promoted. The mechanism may be additionally boosted by high increase in stress, personnel exhaustion, large number of patients in hospital wards, and expectation of individual approach to each patient [54]. However, a high level of agreeableness with simultaneous low level of empathic tendencies and self-reflection may positively influence shaping of the attitude towards gaining communicative skills. A nurse is aware of the need to improve these skills and their significant role in patient care. Self-reflection of this kind may be a key factor that launches shaping positive attitude towards gaining discussed competencies. Summing up, it may be said that agreeableness and empathic tendency are based on coping with emotions. It may be concluded that empathy may serve as a work tool for people with high level of agreeableness and negative attitude towards learning and communicative competences. With regard to the above, it may be assumed that by boosting the study of empathic tendency, the study of coping with emotions may be brought to creating positive attitude towards acquiring communicative competences. Environmental enquiry and discussion with working nursing personnel confirm this view. Nurses participating in the discussion repeatedly stressed the need to introduce courses in the scope of interpersonal skills.

In the model presented above, the positive correlation between agreeableness and professional experience was also recognized. This stems from the fact that agreeableness allows the acquisition of adaptive mechanisms, i.e., the already mentioned escape mechanism, which allows a person to stay in a workplace longer. However, it does not influence the measures taken or attitudes presented by a nurse. There are a plethora of examples from daily life in a nurse’s professional work, where people presenting the syndromes of burnout persist in their professions, and their actions affect new nurses negatively. Based on the assumption above, the function of the mentor in a workplace may turn out to be of key importance (e.g., nurse-to-nurse mentoring [55]), i.e., a person introducing another person to the profession, who, through highly developed empathic tendency, not always supported by developed communicative skills, may shape and create a new generation of nurses in a positive manner. The nurse-to-nurse mentoring [55] may turn out to be an efficient method of reinforcing soft competencies in nurses, which may complement the developed model of factors determining attitudes toward acquiring communicative competences in nurses. Moreover, a mentor may recognize a nurse’s personality better and select an appropriate method of shaping a positive attitude towards acquiring communication skills [56]. The elaborated model creates such a possibility as it is based on both environmental factors and personal ones. It is the mentor who may decide which of those factors should be addressed first.

The developed model, which underwent empirical verification, may also constitute a practical solution for the lack of adequate communicative competences among medical personnel, including nurses, frequently stated in the literature [2,57,58]. Practical implications of the elaborated model should be approached cautiously as the collected data come from cross-sectional research. They have not been verified using observation or linear studies and consequently, require further verification. Despite this obstacle related to interpretation in the context of implementation in the workplace, it may be assumed that this model will help to improve nursing care. In particular, research has shown that negative attitude of medical personnel towards gaining communicative skills is noticeable from patients’ viewpoint. It is reflected in the quality of contact with the patient (e.g., their reluctance to talk and devote their time) [10,59,60]. The key factor in stopping this phenomenon, is empowering medical personnel and people who do not yet work in the profession (students) with tools such as soft skills. However, in the case of professionally active nurses, the task of creating attitudes at the initial stage of education is already impossible. Therefore, for nurses with too high or too low a level of agreeableness, other methods of impact need to be introduced. For instance, activities may be introduced, aiming at strengthening the skills of coping with one’s own emotions, and identification of own self-reflection and patient’s emotions and those of their family [19]. In this regard, strengthening empathic tendency and developing high levels of emotional intelligence [5,26] may bear high significance. The model obtained partially confirms this, as it stresses a positive correlation between agreeableness and empathic tendency and another one between empathic tendency and attitude towards learning communicative competences. What follows, as a result of reinforcement and developing the empathic attitude in nursing personnel, may be positive attitudes towards learning communicative skills.

### Limitations

Interpretation and interference based on the results obtained are subject to certain limitations which need to be taken into account in the process of evaluation. The study group consisted of persons participating in state specialization exams in the field of nursing. It is assumed that these persons represent part of the nursing environment which invests in their professional development both in the areas of knowledge and competences. Therefore, the respondents may prove to show more positive attitude towards learning competences and professional development than persons not participating in trainings and taking postgraduate courses.

The authors of the study did not have any influence on whether participants in the study had any previous training in the field of interpersonal communication or emotional intelligence. Moreover, the authors did not control the intensity of burnout or stress in the workplace. Furthermore, in the study the full personality profile of the participants was not taken into account, and only selected attributes were studied.

Despite the limitations mentioned, the model developed showed an innovative approach to the issue of creating positive attitude towards learning communicative competences. It was proved that a personality trait such as agreeableness, if properly trained in the field of soft skills, may positively contribute to developing an empathic tendency and attitude towards learning communicative competences, as well as contributing to nurses remaining in their profession.

## 5. Conclusions

The elaborated model showed that shaping a positive attitude towards acquiring communication skills is a complex process which involves internal factors (e.g., personality traits, empathic tendencies) as well as external ones (e.g., related to the environment, for instance seniority). As a result, shaping a positive attitude towards acquiring communication skills may be multi-dimensional and individually tailored to the nurses’ needs. It should be stressed that such an influence may be achieved through mentoring and favorable working conditions. However, it needs to undergo empirical verification.

## Figures and Tables

**Figure 1 ijerph-18-01544-f001:**
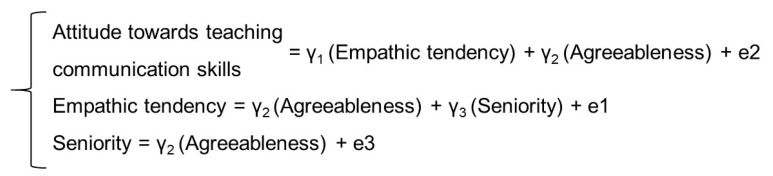
Formal description of the model tested (γ1, γ2, γ3—path coefficients; e1, e2, e3—unobserved variables).

**Figure 2 ijerph-18-01544-f002:**
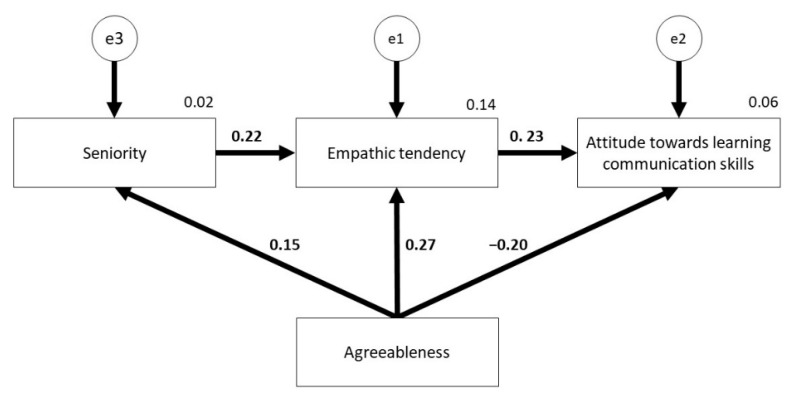
A model of standardized regression weights and squared multiple correlations for measurement.

**Table 1 ijerph-18-01544-t001:** Sociodemographic characteristics of the study sample.

	N	%
Gender		
Female	970	96.0
Male	40	4.0
Place of Residence		
Countryside	254	25.1
Village (population up to 50 thousand)	211	20.9
Small town (51–200 thousand inhabitants)	224	22.2
Large town (201–500 thousand inhabitants)	144	14.3
City > 500 thousand inhabitants	171	16.9
Missing data	6	0.6
Education		
Secondary medical	195	19.3
Bachelor’s degree	306	30.3
Master’s degree	504	49.9
Missing data	5	0.5
Specialization		
Anesthesiological nursing	286	28.3
Internal medicine nursing	141	14.0
Surgical nursing	123	12.2
Geriatric nursing	109	10.8
Oncological nursing	107	10.6
Operating room nursing	83	8.2
Cardiological nursing	44	4.4
Psychiatric nursing	34	3.4
Long-term care nursing	29	2.9
Palliative care nursing	11	1.1
Other	43	4.3

**Table 2 ijerph-18-01544-t002:** Descriptive statistics parameters.

Variable	M	SD	Min	Max	Skew	Kurtosis
Attitude towards learning communication skills ^1^	5.4	1.63	1.0	10.0	0.56	0.77
Empathic tendency ^1^	5.5	1.89	1.0	10.0	0.03	−0.42
Agreeableness ^1^	5.4	1.95	1.0	10.0	0.16	−0.34
Seniority	18.5	9.04	3.0	40.0	0.00	−1.15

M—mean, SD—standard deviation. ^1^ standard ten scores.

**Table 3 ijerph-18-01544-t003:** Likelihood Ratio Chi-Square.

Model	NPAR	CMIN	DF	*p*-Value	CMIN/DF	FMIN
Default	9	0.278	1	0.598	0.278	0.000
Saturated	10	0.000	0			0.000
Independence	4	245.779	6	0.000	40.963	0.244

NPAR—number of parameters, CMIN—chi-square statistics, DF—degrees of freedom, CMIN.DF—normal chi-square, FMIN—minimum fit function.

**Table 4 ijerph-18-01544-t004:** Root Mean Square Error of Approximation.

Model	F0	−90%CI	+90%CI	RMSEA	−90%CI	+90%CI
Default	0.000	0.000	0.005	0.000	0.000	0.067
Saturated	0.000	0.000	0.000	-	-	-
Independence	0.238	0.190	0.292	0.199	0.178	0.221

F0—discrepancy function, RMSEA—Root Mean Square Error of Approximation, CI—confidence interval.

**Table 5 ijerph-18-01544-t005:** Baseline comparison.

Model	NFI	IFI	TLI	CFI
Default	0.999	0.993	1.000	1.000
Saturated	1.000	-	1.000	1.000
Independence	0.000	0.000	0.000	0.000

NFI—Normed Fit Index, IFI—Incremental Fit Index, TLI—Tukey-Lewis index, CFI—Comparative Fit Index.

**Table 6 ijerph-18-01544-t006:** Standardized and unstandardized regression weights.

Construct	Standardized Regression Weights	Unstandardized Regression Weights	SE	CR	*p*-Value
Agreeableness → Seniority	0.151	0.701	0.144	4.855	0.000
Seniority → Empathic tendency	0.224	0.047	0.006	7.596	0.000
Agreeableness → Empathic tendency	0.271	0.264	0.029	9.198	0.000
Empathic tendency → Attitude towards teaching communication skills	0.228	0.196	0.027	7.128	0.000
Agreeableness → Attitude towards teaching communication skills	−0.202	−0.169	0.027	−6.321	0.000

SE—standard error, CR—critical ratio.

## Data Availability

Derived data supporting the findings of this study are available from the corresponding author (M.P.) on request.

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
