# Peer review of "A Model of Factors Determining Nurses’ Attitudes towards Learning Communicative Competences"

_ijerph, 2021, doi:10.3390/ijerph18041544_

Round 1

Reviewer 1 Report

The authors have done a great job. Their work is commendable. However, when reading the article, a number of comments and questions arise.

1. In the introduction, the authors poorly presented the rationale for the study.

1.1. It is not clear from the article why seniority, empathy, and condescension were chosen as study factors. Although the authors themselves say that attitudes towards learning are influenced by external factors (for example, the work environment).

1.2. The authors emphasize the connection of the studied variables with professional activity or communication skills. They do not pay enough attention to the associations of the studied variables with attitudes towards teaching communication skills.

1.3. The authors point to a contradiction in the data on the ratio of work experience with the attitudes of specialists. But they do not explain how these discrepancies are explained.

2. In the tables and in Figure 1, the variable "Attitude towards teaching communication skills" is replaced by "Communication". This misleads the reader.

3. When describing the methods, the authors indicate that the questionnaires were anonymous and after collecting the data it was not possible to identify the personal data of the participant who filled out the questionnaire. But they immediately state that they had access to the database of the Center for Continuing Education for Nurses and Midwives in Warsaw, Poland. This raises doubts as to whether the survey was indeed anonymous.

4. Table 2 provides data on transcripts. And when describing the methods, it is indicated that a five-step Likert scale is used. This makes it difficult to understand the data processing procedure and compare it with the results of other studies.

5. In the discussion section, the authors pay little attention to a theoretical explanation of the revealed patterns. They focus on the role of the proposed model in improving communication skills. However, they partially overestimate this role. For example, the statement that "improvement of communication skills of this kind is possible only with a previously formed positive attitude of the nurse towards the acquisition of these skills, which is preceded by self-reflection." It seems that improvement is possible with a negative attitude, but with a significant external stimulus to learn.

6. Only 44% of the links used by the authors refer to 2015-2020. It seems that the problem under study has recently been little studied, which is not true.

Reviewer 2 Report

This paper aims to evaluate factors that determine the attitude towards learning communicative competencies, and also develop a model based on questionnaire results from a randomly selected sample of nurses from Poland that have taken part in a postgraduate course.  The main model of communication, empathic tendency and agreeableness, and its relationship also with seniority was tested, of which a regression analysis was performed. 

I have the following comments regarding the paper:- 

Article title should be: - A model of factors determining the attitude towards learning communicative competencies of nurses (not "the" nurses)

Within the introduction, it would help if the authors could explain what their definition of communicative skills is in the context of the nursing competency

Discussion: - 

In lines 363 -365, how does the work environment become highly important and relate to professional experience?  

It is difficult to look at how the authors in the discussion section have drawn discussion points and tried to relate the model tested (mainly using quantitative measures), and translated into application in the workplace, e.g. in relation to the work environment.  

It may be better in the discussion to summarise in the first paragraph, what were the author's study findings based on the model testing, then relate it to the impact and relation to other studies/clinical practice.  

In terms of the conclusion, I am finding it difficult on how the model relates or suggests for shaping competencies and acquiring methods of coping with difficult situations in a hospital ward, which is what a reader will find difficult.

Reviewer 3 Report

The article is interesting and discusses an important topic.

Introduction

After reading the entire article, some parts of the introduction need to be moved to the discussion for better comparability of results with the published literature. This will shorten the introduction and make it more to the points to allow a more focused paragraph.

Materials and method

No comments

Results

No comments

Discussion:

Needs extensive editing.

The discussion is relevant but somewhat ambiguous. The authors are highlighting the importance of empathic tendency, personality traits, work environment, past experience, and continuous education on acquiring communicative competencies. However, these factors are not clearly delineated to explain how they are shaping the nurse's attitude. The same ideas are being repeated sparingly which creates ambiguity. I propose to review and simplify the discussion part to convey a clear and cohesive flow in the writing.

-The enrolled male nurses is limited, does the gender differences affect the influence of different factors that shape nurses attitude in acquiring communicative competences?

364: review the following sentence

“Therefore work environment becomes highly important, as it conditions work environment”.

Limitations

Needs extensive Editing

450-451: the following sentence is not clear:

“What also must not go unnoticed is the stress factor, which was natural given the time of filling in the tool”.

451-452: the following sentence is not clear, can the author elaborate more.

“The persons took a state exam which granted them the title of specialist in the field of nursing”.

Conclusion:

The conclusion should be reviewed, rephrased, and shortened. Same repeated idea.

- 465, 470, And 472: Shaping a positive attitude towards acquiring communicative skills

-472-474: the sentence needs a verb

Round 2

Reviewer 2 Report

Thank you for addressing the reviewer comments in the revised manuscript with changes highlighted. I have no further comments.